# Managerial thinking in neonatal care: a qualitative study of place of care decision-making for preterm babies born at 27–31 weeks gestation in England

Caroline Cupit ,[1,2] Alexis Paton ,[3] Elaine Boyle,[1,4] Thillagavathie Pillay ,[4,5] Natalie Armstrong ,[1] For the OPTI-PREM Study Team

CC and AP are joint first authors.

For numbered affiliations see end of article.

**Correspondence to**
Dr Caroline Cupit;
caroline.cupit@le.ac.uk

## ABSTRACT

**Objectives** Preterm babies born between 27 and 31 weeks of gestation in England are usually born and cared for in either a neonatal intensive care unit or a local neonatal unit—with such units forming part of Operational Delivery Networks. As part of a national project seeking to optimise service delivery for this group of babies (OPTI-PREM), we undertook qualitative research to better understand how decisions about place of birth and care are made and operationalised.

**Design** Qualitative analysis of ethnographic observation data in neonatal units and semi-structured interviews with neonatal staff.

**Setting** Six neonatal units across two neonatal networks in England. Two were neonatal intensive care units and four were local neonatal units.

**Participants** Clinical staff (n=15) working in neonatal units, and people present in neonatal units during periods of observation.

**Results** In the context of real-world neonatal practice, with multiple (and rapidly-evolving) uncertainties relating to mothers, babies and unit/network capacity, 'best place of care' protocols were only one element of much more complex decision-making processes. Staff often made judgements from a less-than-ideal starting point, and were forced to respond to evolving clinical and organisational factors. In particular, we report that managerial considerations relating to demand and capacity organised decision-making; demand and capacity management was time-consuming and generated various pressures on families, and tensions between staff.

**Conclusions** Researchers and policymakers should take account of the organisational context within which place of care decisions are made. The dominance of demand and capacity management considerations is likely to limit the impact of other improvement interventions, such as initiatives to integrate families into the neonatal care provision. Demand and capacity management is an important element of neonatal care that may be overlooked, but significantly organises how care is delivered.

## STRENGTHS AND LIMITATIONS OF THIS STUDY

⇒ This study is the first to use qualitative methods to explore practices relating to 'best place of care' using data from real-world neonatal practice.
⇒ We undertook extensive observation (280 hours) in six neonatal care units (neonatal intensive care units and local neonatal units), and captured the perspectives of a range of neonatal staff (doctors, nurses, transport specialists).
⇒ We interviewed only a few staff from each unit, thus limiting the analytic 'depth' and comparison between sites.
⇒ Our data did not include observations/interviews in obstetrics or maternity care, or with neonatal transport services, neonatal network managers or cot locating facilities and we therefore do not discuss the operational arrangements and interdependencies between these services.

## INTRODUCTION

Babies needing medical care after birth are cared for in three types of neonatal units across England, organised into Operational Delivery Networks: neonatal intensive care units (NICU), local neonatal units (LNU) and special care baby units. Evidence suggests that outcomes are better when extremely preterm babies (≤26 weeks of gestation) are born in a maternity service attached to an NICU.[1 2] For the next most vulnerable group of babies (ie, those born between 27 and 31 weeks of gestation (hereafter referred to as born at 27–31 weeks)), there is little evidence to guide decisions about their place of birth and care. More evidence is available for babies born >31 weeks.[3] Babies born at 27–31 weeks represent around 12% of all preterm births each year in England, use twice as many neonatal bed days/year compared with

BMJ

extremely preterm babies, and account for over a third of all neonatal care days nationally.[4]

In the absence of clear evidence to guide optimal place of care, babies born at 27–31 weeks are born and cared for across both NICU and LNU. A national project seeking to optimise service delivery for this group of babies, OPTI-PREM,[4] is investigating whether new recommendations should be developed for subgroups of babies within the 27–31 week range, based on mortality or morbidity analyses. (Detailed criteria regarding weights and multiple babies at the lower end of the gestational age have been modified since this study was carried out.)[5] In practice, some women (who are known to be at high risk of complicated preterm delivery) may be transferred to a specialist unit, but many others continue to deliver in a maternity service linked to an LNU. Some of these may be transferred post-delivery to an NICU due to their degree of illness or prematurity.[5] These ex-utero transfers are known to create particular stresses for babies and families[6–9] and should be avoided where possible.[10] In the past, transfers have been shown to be related to regional demand and capacity issues.[11–13] However, a high number of transfers have continued despite neonatal service reorganisation.[14] There is little recent evidence on how decisions about place of care are made in practice.

This article draws on qualitative work exploring neonatal care for babies born 27–31 weeks, undertaken as part of the OPTI-PREM study. Using observation in neonatal units and interviews with staff, we investigate how decisions about place of care are made and operationalised for this group of babies. We are aware of no previous ethnographic work addressing place of care decision-making in this clinical context.

## METHODS

We undertook an ethnographic study in two Neonatal Operational Delivery Networks in England. Within each network, an NICU and two attached LNU were included in the study (six neonatal units in total). From January to October 2018, observations and staff interviews were conducted in order to explore decision-making about place of care in a real-world context.[15–17] The wider study also captured experiences of parents but as parents were not directly involved in making decisions about babies' place of care, these data will be reported separately.

Sites were identified and invited to participate through the study principal investigator's professional network. Staff working in the units were introduced to the study by members of the OPTI-PREM project team. Written information on the project was provided. Observations were conducted by AP (an experienced qualitative researcher), and included ward rounds, daily clinical activities, bedside discussions with parents, transfer discussions and referrals and transfers themselves. Observations were guided by an observation framework (online supplemental file 1). Verbal consent for observations was sought from all staff and parents in the unit prior to each period of observation. Anonymised field notes were made. AP proactively responded to sensitive situations. Every fieldwork day, AP checked-in with staff to identify any babies and families in a particularly challenging situation (eg, palliative care) and did not disturb these families. In emergency situations, or when it seemed otherwise appropriate, AP withdrew from the bedside area.

As well as undertaking informal discussions with staff and parents during ethnographic data collection, AP also conducted formal interviews with staff. All doctors and nurses working at the units on observation days were approached for interview. Participation was voluntary. Fifteen staff agreed to be formally interviewed (many more contributed to the ethnographic element of the study, through being observed and via informal chats), and written consent was obtained. Most non-participation was due to challenges with scheduling around shifts. Interviews were semi-structured based on a topic guide (online supplemental file 2) developed through literature review and discussions within the project team. Interviews were conducted by AP on-site, lasted up to an hour and were audio recorded, transcribed verbatim and de-identified.

Analysis was inductive and interpretative, with AP using the constant comparative method to undertake initial coding.[18] CC also carried out detailed reading of transcripts/field notes, and employed an analytic approach known as 'institutional ethnography'[19–22] to study place of care decision-making as part of socially-organised work processes.[23–26] NVivo software was used to organise and retrieve data.

### Patient and public involvement

The OPTI-PREM project was supported by a Bliss[27] volunteer parent panel, which was involved in designing and overseeing the research.

### RESULTS

In total, 280 hours of observations were conducted and 15 members of staff were interviewed across the six units. A breakdown of professional roles is shown in table 1.

The findings reported here are presented in two main themes. The first theme highlights the complex, multifactorial context in which staff made decisions about place of care and consequently the difficulties of producing and applying 'best place of care' protocols, even for *subgroups* of babies within the 27–31 week range. The second theme specifically focuses on how staff integrated 'managerial thinking' about demand and capacity management (within the network's contractual framework) into clinical decision-making about the 'best place of care' for each individual baby.

### 'Best place of care' protocols within contextualised decision-making processes

Place of birth and care was to a large extent governed by regional protocols, which determined what should happen in the case of an expected preterm birth at 27–31

**Table 1** Participant overview

| Site | Designation | Clinician type | Number of participants (Total=15) |
|---|---|---|---|
| Site 1 | NICU | Doctor (consultant) | 1 |
| Site 2 | NICU | Doctor | 1 |
| | | Nurse | 1 |
| Site 3 | LNU | Doctor (registrar) | 1 |
| | | Nurse | 2 |
| Site 4 | LNU | Doctor (consultant) | 1 |
| | | Nurse | 1 |
| Site 5 | LNU | Doctor (consultant) | 1 |
| Site 6 | LNU | Doctor (consultant) | 1 |
| | | Nurse | 2 |
| Working across sites | | Transport nurse | 3 |

LNU, local neonatal units ; NICU, neonatal intensive care units .

weeks gestation.[5 28 29] For example, regional care pathways meant that LNU should only accept and care for babies born at ≥27/28 weeks, depending on the unit. For expectant preterm births in earlier gestational subgroups, the woman (with baby in utero) should be transferred to a maternity service attached to an NICU, assuming mother and fetus were considered adequately stable. Otherwise, the baby could be transferred post-delivery to an NICU for ongoing care if needed. Neonatal staff looked to such protocols (along with dynamic assessment of unit/network resources) to determine how to proceed.

> Majority of the time, it's quite a simple decision. For example, guidelines [say] we can't deliver, or we shouldn't manage anyone less than 27 weeks here […] And [we] have clear guidelines of how many nurses we need to run an ITU [intensive therapy unit], how much space you need and all that.
>
> Doctor12, LNU

Although, as indicated above, these decisions were often simple *in principle*, they involved *in practice* multiple uncertainties relating to both mother and baby's condition. Staff knew that, in an idealised scenario, the safest possible option would be for all mothers to have access to the specialist resources of a maternity service linked to an NICU, but they also recognised that this level of specialist care was unnecessary for some babies. The problem was that, although some information was known prior to delivery (and an informed guess made about a baby's subsequent condition), a baby's post-delivery condition was unpredictable:

> These are [a] very notorious group because [babies] can behave like an extreme premature, or they can just need a bit of support […]. It's very difficult to predict these babies.
>
> Doctor12, LNU

Pre-delivery, it was difficult to predict which babies could be safely managed on LNU, and decisions were reliant on local expertise. Staff considered this critical decision-making expertise to be lacking in some LNU:

> We can scan but we don't have the expertise to confidently say there's no bleeds going on, or the head's normal. So small things add up to not having good care or optimal care for the extreme premature baby. [For example], we are not funded, we've not got the personnel or the skills, because we don't deal with that so often.
>
> Doctor12, LNU

Staff therefore understood that there was extra risk attached to delivering this group of babies in many LNU (expertise and equipment varied from unit to unit).

Not only were babies unpredictable, but so too were mothers and families. For example, a mother might not travel to the preferred NICU due to imminent labour, or social circumstances. Units therefore had to handle imminent deliveries regardless of whether or not they were equipped to care for the baby according to recognised standards and guidelines for neonatal care. Even mothers classified in advance as being 'high risk' (and therefore booked at an NICU) often presented in labour at their local unit, which was then obliged to manage the delivery:

> Triplets were booked for Site 1 but were born at Site 5. Consultant said this happens a lot—that high risk pregnancies are booked at Site 1 but then Mum presents at Site 5 too far along to be moved.
>
> Observation, Site 5, LNU

Often, a suboptimal delivery at an LNU was not due to the unpredictability of mother or baby, but due to capacity issues in local NICU or lack of availability of transport teams (that were needed to safely carry out some in-utero transfers). Units often ended up accepting babies for which they did not have appropriate capacity (eg, equipment/staffing):

> [The nurse said:] Sometimes [the LNU] is full and [they] tell people to rock up [arrive late, without warning] without checking [we] can take the pressure.
>
> Observation, Site 5, LNU

A unit's own capacity management had to be undertaken in relation to capacity issues across the network/region. As the excerpt above indicates, units were not always aware of/responsive to other units' management concerns and this was exacerbated by variation in policies and fluctuating capacity across units of the same designation:

> […] every hospital is different. Policies are different. Maybe it will be in time standardised within the NHS.
>
> Nurse10, NICU

As highlighted above, many of the 27–31 week babies in this study were born in LNU—and this included those who did not meet established local criteria. (See Edwards and Impey[30] who found that, even in the <27 week group of extremely preterm babies, only 50% were actually born in a specialist unit, despite this being the recommendation.) In practice, protocols were frequently contravened due to limited capacity across obstetrics/maternity services and NICU, along with the other factors relating to mother and baby that we have already outlined. Other babies were born in NICU (eg, due to the mother's home being in close proximity) even though they could be adequately cared for on an LNU.

In practice, therefore, many place of care decisions were not taken from a stable, pre-clinical position, but within an existing clinical context (either LNU or NICU)—in which care for mother and baby was already underway and rapidly evolving:

> If the birth is imminent then we just go for it. Personally I've dealt with babies who are twenty-three-and-a-half weeks onwards [at another unit]. So personally speaking I'm comfortable, and I'm pretty sure my nursing colleagues are pretty comfortable. So we deal with them, but as soon as the baby comes out, one of my colleagues starts looking around hunting for a bed.
>
> Doctor20, LNU

In such situations, staff had to cope with whatever happened and prepare to transfer the baby (if required) after delivery.

In other circumstances, staff made calculated judgements to keep mothers at a local hospital, based on their experience and dynamic knowledge of multiple contextual factors:

> On a December cold, wintery night and a mum came in-utero [with triplets]. And there was possibly a small window for her to go elsewhere, but I carefully calculated the risk and said "your twenty-nine-weekers have very good growth for all those babies and, it's winter and there is a chance that things might happen en route…". At the time I had a very experienced registrar with me and I did have another member of the team who could definitely come back [into work after leaving shift]. And this mum had had steroids as well in the last week. So looking at all those calculations I actually did look after them.
>
> Doctor30, LNU

Assessment of the best place of care was often re-evaluated following delivery, with some decisions being relatively straightforward (in principle at least)—based on urgent clinical needs—while others required more complex assessment of the pros and cons within the wider unit/network context.

When staff made decisions (with baby either in utero or ex utero), they not only applied protocols but also attended to a huge range of uncertainties and a rapidly-evolving situation relating to mother, baby and also unit/network capacity, which affected how those protocols were applied. Family circumstances and preferences were sometimes accommodated in decision-making (eg, when mother and baby were both inpatients, efforts were made to locate them at the same hospital), but decisions were largely determined by the clinical needs of the baby, organised within a wider demand and capacity management framework—as discussed below.

### 'Thinking managerially' about where individual babies should be born and cared for

Decisions about place of care for most of the babies born at 27–31 weeks was highly contextualised and involved assessment of the particular needs of individual babies. It was clear that all staff aspired to the best quality care for individual babies. However, each individual baby's clinical assessment was made within a socially-organised clinical context. Our data show that resources (and their distribution across networks) significantly shaped place of care decisions about individual babies born at 27–31 weeks—with specialist maternity services and NICU only being available in city locations, and without the necessary capacity to accommodate all 27–31 week births.

More specifically, we found that a large amount of staff work was dedicated to matching the (either anticipated or known) clinical needs of the neonatal baby population ('demand') with the availability of expertise and technologies ('capacity')—both of which were scattered geographically across a region, and were constantly being reconfigured. Staff drew not only on knowledge of the baby's needs (eg, requirement for ventilation, tube feeding, surgery), but also on knowledge of capacity and resources within their own unit within the context of the wider network:

> We like to think managerially as well [as clinically]. If [another unit with limited specialist capacity] has a twenty-eight week [baby] and they ask whether we would like to accept, we are more than happy. Because, not only it's good for our practice that we keep on getting prem babies, it [also] brings money to the [hospital]. So we are always looking for kids…
>
> Doctor20, LNU

Staff spoke of their 'network responsibility' to try to accommodate babies—a responsibility linked to national tariffs for different categories of care[31 32] and locally negotiated contracts and payments.[33] They knew that, in relation to this 'responsibility', they needed to 'think managerially' about capacity and resources. Such thinking (and the associated decision-making) was concerned with the unit's need to maximise income, while avoiding 'unsafe' practice (as determined by capacity/staffing protocols[10]). In the excerpt above, one unit was short of the required capacity, while another was looking to ensure that available capacity was generating income.

Staff were locked into continuous work to review their own capacity and respond to the requests of other units within the network. 'Capacity' in this context included many different elements—including staff mix, qualifications, experience, space and technologies:

So in our [LNU], [we can take] twenty eight weeks and above. And then we look at ourselves. How are we? Is there enough nursing staff? Are there enough doctors? And then we look at bed capacity, and [intensive care capacity], whether there's enough space. […] Sometimes it happens that we shift babies out because of capacity.

Doctor20, LNU

All units logged requests for transfers from other units, noting why they accepted or refused. One transport nurse emphasised that units did their best to accommodate babies who were vulnerable to deterioration as a consequence of transfer:

[Units] try and shuffle to make [babies] fit in.

TransportNurse05A

However, in all six units, staff or cot capacity were the most often cited reasons for a refusal.

Units were paid for their 'activity' (number of babies), so accepting new babies was important for income generation as well as to maintain the unit's status (also linked to the category of babies they were allowed to accept):

You have to accept, because we are an LNU, we cannot afford our activity to go down. Because [then] we'll be relegated to a [non-specialist] unit. So we won't want to be seen as refusing […] We can get deskilled […] and the more activity we do, the more funds we get [so] it's a win-win situation.

Doctor26, LNU

LNU therefore tried to retrieve babies who had been transferred to NICU, as soon as they were clinically ready for repatriation:

There is a baby here from an out-of-network hospital. [The hospital] has been ringing to see if it can come back.

Observation, Site 1, NICU

In addition, each baby attracted a different tariff, according to the category of care that was required[31 32]—and this was an important consideration:

While waiting, [another hospital] has rung up with an ex-utero transfer (twins). "We must be the only place with beds!" Everyone is excited to take the babies as it is an ITU transfer and that brings in a lot of money. [Staff member] singing "one thousand and twenty-one pounds. Thank you!

Observation, Site 3, LNU

However, maximising capacity, and meeting the needs of very sick babies, was sometimes in tension with providing a broad spectrum of neonatal delivery for the local population. For example, babies who were clinically stable, and whose local unit was an NICU, were often transferred out, sometimes to an inconvenient location, in order to accommodate babies with more complex needs:

The units are very good; if it is a really sick baby that's got a specific problem then they're very good at trying to make beds, so often we'll move other babies out … we do jiggling around quite a lot.

TransportNurse05

So it's finding a divide between running a hospital (just purely thinking about the service and how much activity you can churn through at the high end of intensive care) vs running a service that makes sure you provide the right service for the families of your local population.

Doctor06, NICU

Unlike decisions that were made based purely on clinical need, decisions orientated entirely to demand and capacity management took an emotional toll on staff:

I think the hardest conversation is the move out for capacity reasons. So making a decision to impact significantly on a family […] for the care of a different baby who they don't have nothing to do with can be quite difficult sometimes.

Doctor09, NICU

Capacity-related issues could also create tension between staff groups:

The doctor was complaining about how there is no standardised practice for closing the unit […] The nurses always want to operate at a gold standard, but it is impossible to operate at that level because the unit is always falling short due to staff shortages.

Observation, Site 1, NICU

Senior staff were drawn into extensive work to account for capacity-related decisions, and they passed this pressure onto staff within their units:

Consultants want to have a better way of recording why babies can't be accepted [for upwards reporting]. They want specifics, so they were saying things like "Why have you said we're busy? What is it about the category of babies in today that means we can't take another one?

Observation, Site 1, NICU

In practice, 'clinical' and 'managerial' ways of thinking were entirely intertwined within the work of neonatal staff, including their decision-making about place of care.

## DISCUSSION

This paper provides empirical evidence of how 'best place of care' protocols are incorporated into practical, real-world decision-making for babies born between 27 and 31 weeks of gestation in England. Our findings highlight that clinical decisions about the 'best place of care' for individual babies are, in practice, and to a significant extent, orientated to demand and capacity management, as part of a unit's contractual arrangements within the network. We have drawn attention to the managerial work that is involved in staff decision-making about place of care. When staff make these decisions for an individual baby, they integrate clinical knowledge about that baby's needs with management understandings of what is 'best'. We have highlighted that neonatal staff are highly skilled at making complex judgements about place of care based on their in-situ knowledge about the needs of babies, and the resources (eg, staffing, equipment, cots) that are available to meet those needs.

Although sometimes it appears that 'clinical' and 'managerial' ways of knowing are aligned (and appear to present no difficulties for staff, parents or babies), at other times they come into tension. Institutional systems and processes relating to demand and capacity management (contractual funding arrangements, safe staffing protocols etc) often create difficulties as they coordinate neonatal work. Overall, a large amount of staffing resource is involved in negotiating between babies' individual needs and units' resources. Staff spend considerable energies balancing the competing needs of babies (whether they are transferred or stay put in their birth hospital), within the constantly-reconfiguring distribution of resources within a network. This demand and capacity management work inevitably diverts staff from other activities, and impacts morale, as well as their time to support babies and families. This is likely to have a negative impact on important partnership-working with parents.[34] Parents may also be faced with considerable knock-on challenges in order to maintain their active parenting role, as demand and capacity considerations take precedence and mother and baby are 'juggled' between units. Our findings align with studies in Canada, which have made visible the considerable parental work involved in neonatal care (eg, in transfers, Family Integrated Care) and, importantly, have highlighted dissonance with major protocol-driven institutional systems, which fail to take account of this work.[9 35] Our study contributes to a small corpus of work which can support stakeholders understandings of how contractual and funding configurations shape clinical decision-making (and impact on care) in practice.[36] This institutional context, and analysis of the practical activities involved, have frequently been absent from research on neonatal place of care and transfers.

Our findings are highly relevant to stakeholders who see solutions in standardised protocols and pathways to deliver improvements. We found that place of care protocols were only one element within much more complex decision-making processes. This observation has implications for both the production and application of evidence relating to 'best place of care'. In relation to *production* of evidence, studies attempting to assess the clinical outcomes of babies born on NICU compared with those born on LNU are likely to be challenging as NICU and LNU vary considerably within these classifications, and the characteristics of babies born in each are not standardised. Transfers between units are also likely to make comparisons extremely difficult. In relation to *application* of evidence, standardised protocols for this group of babies may also present difficulties within a context that involves un-standardised babies, mothers, geographies, resourcing and so on. Even if future evidence indicates that some subgroups of babies in the 27–31 week range are (theoretically) best born and/or cared for in hospitals with an NICU, such highly specialist and costly facilities will not be on the doorstep of all mothers and will undoubtedly have limited capacity. Stakeholders should be alert to the situatedness of decision-making and the potential for unintended consequences following the introduction of new protocols.[37]

There are some limitations to this study. Data are based on a sample of neonatal units that were purposively selected for the project. Within units, data collection did not include observations or interviews in obstetrics or maternity care, or with neonatal transport services, neonatal network managers or cot locating facilities and we therefore do not discuss the operational arrangements and interdependencies between these services and neonatal units. Future work might look in more detail at this intersection. Additionally, only a few staff from each unit were interviewed formally, thus limiting the analytic 'depth' and comparison between sites. Nevertheless, our findings provide important insights into the real-world context in which place of care decisions are made and operationalised.

## CONCLUSIONS

Researchers and policymakers should take account of the organisational context in which place of care decisions for preterm babies (eg, born 27–31 weeks) are made. Although some standardised guidelines are likely to be useful, it will continue to be necessary for staff to make on-the-spot, skilled judgements about place of care. It is important that this skilled work is recognised in workforce and wider neonatal care planning. Notably, this study found that place of care decisions are significantly orientated to demand and capacity management—alongside other, more obvious, clinical considerations. In practice, the dominance of these management considerations is likely to limit the impact of other improvement interventions, such as initiatives to integrate families into the neonatal care provision. Demand and capacity management is an important element of neonatal care that may be overlooked, but significantly organises how care is delivered.

**Author affiliations**
[1]Department of Health Sciences, University of Leicester, Leicester, UK
[2]Nuffield Department of Primary Care Health Sciences, University of Oxford, Oxford, UK
[3]Sociology and Policy, Aston Medical School, Aston University, Birmingham, UK
[4]Neonatology, University Hospitals of Leicester NHS Trust, Leicester, UK
[5]Research Institute for Health Related Sciences, University of Wolverhampton, Wolverhampton, UK

**Acknowledgements** The authors thank all the staff whose experiences are reported in this article, and the other participants involved in the OPTI-PREM study. We are also grateful to the reviewers, especially Catherine Ringham, for their helpful comments. NA is supported by a Health Foundation Improvement Science Fellowship and also by the National Institute for Health & Care Research (NIHR) Applied Research Collaboration East Midlands (ARC EM). The views expressed are those of the authors and not necessarily those of the NHS, the NIHR or the Department of Health and Social Care. The OPTI-PREM study team: Natalie Armstrong; Victor L Banda; Vasiliki Bountziouka; Caroline Cupit; Kelvin Dawson; Elaine M Boyle; Elizabeth S Draper; Abdul Qader T Ismail; Bradley Manktelow; Neena Modi; Alexis Paton; Oliver Rivero-Arias; Sarah E Seaton; Miaoqing Yang; and Thillagavathie Pillay (Chief Investigator).

**Collaborators** For the OPTI-PREM Study Team: Victor L Banda; Vasiliki Bountziouka; Kelvin Dawson; Elizabeth S Draper; Abdul Qader T Ismail; Bradley Manktelow; Neena Modi; Oliver Rivero-Arias; Sarah E Seaton; Miaoqing Yang.

**Contributors** Obtained funding: TP, EB and NA. Study design: NA. Data analysis: CC and AP. Drafted manuscript: CC and AP. Interpreted to neonatal practice: TP and EB. Conceptualised and critically revised the manuscript: CC. All authors approved the final version. CC and AP contributed equally to this paper. Guarantor: NA.

**Funding** This work was supported by the National Institute for Health & Care Research, Health Services and Delivery Research Stream, Project number 15/70/104.

**Competing interests** None declared.

**Patient and public involvement** Patients and/or the public were involved in the design, or conduct, or reporting, or dissemination plans of this research. Refer to the Methods section for further details.

**Patient consent for publication** Not applicable.

**Ethics approval** This study involves human participants and was approved by North East – Tyne and Wear South REC (IRAS 212304). Participants gave informed consent to participate in the study before taking part.

**Provenance and peer review** Not commissioned; externally peer reviewed.

**Data availability statement** No data are available.

**ORCID iDs**
Caroline Cupit http://orcid.org/0000-0002-3377-8471
Alexis Paton http://orcid.org/0000-0003-4310-6983
Thillagavathie Pillay http://orcid.org/0000-0002-4159-3282
Natalie Armstrong http://orcid.org/0000-0003-4046-0119

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
