## [Reviewer comments · BMJ Open]

ARTICLE DETAILS

TITLE (PROVISIONAL)	Managerial thinking in neonatal care: a qualitative study of place of care decision-making for preterm babies born at 27-31 weeks gestation in England
AUTHORS	Cupit, Caroline; Paton, Alexis; Boyle, Elaine; Pillay, Thillagavathie; Armstrong, Natalie

VERSION 1 – REVIEW

REVIEWER	Ringham, Catherine Thompson Rivers University, Nursing
REVIEW RETURNED	02-Feb-2022

GENERAL COMMENTS	Thank you for the opportunity to review this very well written manuscript. It was a pleasure to read! The institutional/managerial coordination of care is often not addressed nor do researchers highlight the highly complex, in the moment decision-making that goes on, especially in the neonatal environment. You have provided data excerpts that illustrate how the various care providers think and act on written and unspoken “criteria”. This is an important discourse to untangle. You have accomplished this effectively. I recommend you address the following: • Indicate that REB approval was obtained• Discuss limitations to this study• Page 8- A fuller explanation of what AP did to proactively respond to sensitive situations. Did AP remove themselves from the Unit or area when there were challenging situations for staff and/or parents, for example during a resuscitation?• Page 11- Rock up is a phrase that may be unfamiliar to readers outside of the UK. Is it possible to explain?• Page 15- managing available capacity and generating income consistently is a delicate balance. You suggest this process is entirely based on managerial decision-making. Where are the parents needs considered in all of this? (I look forward to reading the manuscript where you focus on parents’ experiences) I recommend this important article for publication.
--

REVIEWER	Tan, Audrey University College London, Office of the Vice-Provost (Research, Innovation & Global Engagement)
REVIEW RETURNED	08-Feb-2022

GENERAL COMMENTS	This was an interesting manuscript in an important field.
---

	The objective of the manuscript is stated at the end of the Introduction section, but there could be confusion with the research question outlined in the first paragraph of the Methods section. I would remove this from the Methods and have the objective clearly stated at the end of the introduction. If the research question 'What are staff...' is that of the wider study, make sure this is clearly explained. Some of the direct quotations from participants are hard to understand, would suggest minor editing so they're easier to read. It would help the reader if the two main themes in the Results section were divided into sub-themes as it is currently difficult to maintain the thread of the argument. Perhaps this is due to the style guidelines of the journal, but I did not find a discussion of the study's limitations in the Discussion section, though it was listed a bullet points after the abstract. If this does not contravene style guidelines, I would recommend sections in the Discussion to discuss strengths and limitations, as well as areas for future research. Would suggest moving the last paragraph of the Results section to the Discussion section. The Discussion section could also be improved by including a discussion of how your results compare to other similar studies in the field - where do your findings align and differ with other studies? The conclusions outlined in the Abstract and in the Conclusion section are not as impactful as they could be. I would suggest revising the conclusion so that it highlights the most important findings and recommendations. One sentence ends 'should take account of this real-world evidence.' But it's unclear what 'this real-world evidence' is. The reader should be able to read the conclusion and be able to know what it's referring to without having to read the previous section.
--	--

VERSION 1 – AUTHOR RESPONSE

Reviewer 1

No	Reviewer comment	Response
1.1	Thank you for the opportunity to review this very well written manuscript. It was a pleasure to read! The institutional/managerial coordination of care is often not addressed nor do researchers highlight the highly complex, in the moment decision-making that goes on, especially in the neonatal environment. You have provided data excerpts that illustrate how the various care providers think and act on written and unspoken "criteria". This is an important discourse to untangle. You have accomplished this effectively.	Thank you very much for your positive feedback and your constructive comments, which have helped us improve this manuscript. Detailed responses are below.

No	Reviewer comment	Response
1.2	I recommend you address the following: Indicate that REB approval was obtained	There is a section for Ethics Approval at the end of the manuscript, which we have completed.
1.3	Discuss limitations to this study	We have added a paragraph addressing strengths and limitations at the end of the Discussion. [See response to E.1.]
1.4	Page 8- A fuller explanation of what AP did to proactively respond to sensitive situations. Did AP remove themselves from the Unit or area when there were challenging situations for staff and/or parents, for example during a resuscitation?	We have added a couple of sentences providing more detail (see paragraph 2 of Methods).
1.5	Page 11- Rock up is a phrase that may be unfamiliar to readers outside of the UK. Is it possible to explain?	Thank you for highlighting this - we have added an explanation.
1.6	Page 15- managing available capacity and generating income consistently is a delicate balance. You suggest this process is entirely based on managerial decision-making. Where are the parents' needs considered in all of this? (I look forward to reading the manuscript where you focus on parents' experiences)	We have added a couple of sentences to address your question (p14). The needs of parents were considered, but these were marginal to other considerations.

Reviewer 2

No	Reviewer comment	Response
2.1	This was an interesting manuscript in an important field.	Thank you for your positive feedback and for your constructive comments below, which have been really useful in improving the manuscript. We address them point by point below.
2.2	The objective of the manuscript is stated at the end of the Introduction section, but there could be confusion with the research question outlined in the first paragraph of the Methods section. I would remove this from the Methods and have the objective clearly stated at the end of the introduction. If the research question 'What are staff...' is that of the wider study, make sure this is clearly explained.	Thank you for pointing out an apparent inconsistency here. We have removed the confusing articulation of the research question in Methods, as recommended.
2.3	Some of the direct quotations from participants are hard to understand, would suggest minor editing so they're easier to read.	We have made minor punctuation and clarification changes to make quotations easier to read.
2.4	It would help the reader if the two main themes in the Results section were divided into sub-themes as it is currently difficult to maintain the thread of the argument.	Thank you for your comment. We appreciate that this is a more narrative style than is currently popular in the presentation of qualitative research. However, we have deliberately used this approach as we think it has some significant benefits for this paper. In particular, we found during the drafting process that it was unhelpful to partition the data into multiple themes, as the excerpts and

No	Reviewer comment	Response
		issues involved were so intricately connected, and fitted better under two broader headings. We worked hard to present a good quality account, within the word limit, by walking the reader through the data using a more narrative approach. Bearing in mind that Reviewer 1 was pleased with the readability and flow of the manuscript, we would prefer to leave the findings in their current form. We will leave the editors to input on this if necessary. We hope that reducing the length of the second findings section (as per 2.6 below) may in part address this comment.
2.5	Perhaps this is due to the style guidelines of the journal, but I did not find a discussion of the study's limitations in the Discussion section, though it was listed a bullet points after the abstract. If this does not contravene style guidelines, I would recommend sections in the Discussion to discuss strengths and limitations, as well as areas for future research.	Thank you. We have added a section on strengths and limitations, and future research, at the end of the Discussion. [See response to E.1.]
2.6	Would suggest moving the last paragraph of the Results section to the Discussion section. The Discussion section could also be improved by including a discussion of how your results compare to other similar studies in the field - where do your findings align and differ with other studies?	We have moved this section as suggested, and made minor changes to the Discussion (e.g. ordering) to ensure a good flow, and avoid repetition, in making this change. We have also added some comparison with similar studies in the field, whilst being mindful of the author guidelines limiting the length of the discussion section.
2.7	The conclusions outlined in the Abstract and in the Conclusion section are not as impactful as they could be. I would suggest revising the conclusion so that it highlights the most important findings and recommendations. One sentence ends 'should take account of this real-world evidence.' But it's unclear what 'this real-world evidence' is. The reader should be able to read the conclusion and be able to know what it's referring to without having to read the previous section.	Thank you. We agree that the Conclusion needed work. We have re-written (in both main text and abstract), and hope that you find this more impactful. We have also made a minor amendment to the title of the article in order to better reflect our findings and provide greater impact. However, we are content to revert to the previous title should the editors recommend this. Thank you for your thoughtful comments, which have been really useful in improving the paper.

VERSION 2 – REVIEW

REVIEWER	Ringham, Catherine Thompson Rivers University, Nursing
REVIEW RETURNED	06-May-2022

GENERAL COMMENTS	The authors have carefully attended to my review comments/questions and responded adequately. The limitations
---

	have been fleshed out more thoroughly in this version. I appreciated how well you strengthened the conclusion as well. Thank you for presenting this important topic in a narrative style that illustrates the interconnections and complexity of the issues you have brought to our attention.
--	--